# The Emerging Role of Immunotherapy in Intrahepatic Cholangiocarcinoma

**DOI:** 10.3390/vaccines9050422

**Published:** 2021-04-22

**Authors:** Oraianthi Fiste, Ioannis Ntanasis-Stathopoulos, Maria Gavriatopoulou, Michalis Liontos, Konstantinos Koutsoukos, Meletios Athanasios Dimopoulos, Flora Zagouri

**Affiliations:** Department of Clinical Therapeutics, School of Medicine, National and Kapodistrian University of Athens, Alexandra Hospital 80 Vasilissis Sophias, 11528 Athens, Greece; johnntanasis@med.uoa.gr (I.N.-S.); mariagabria@gmail.com (M.G.); mliontos@gmail.com (M.L.); koutsoukos.k@gmail.com (K.K.); mdimop@med.uoa.gr (M.A.D.); florazagouri@yahoo.co.uk (F.Z.)

**Keywords:** intrahepatic cholangiocarcinoma, biliary tract cancer, immunotherapy, tumor microenvironment

## Abstract

Biliary tract cancer, and intrahepatic cholangiocarcinoma (iCC) in particular, represents a rather uncommon, highly aggressive malignancy with unfavorable prognosis. Therapeutic options remain scarce, with platinum-based chemotherapy is being considered as the gold standard for the management of advanced disease. Comprehensive molecular profiling of tumor tissue biopsies, utilizing multi-omics approaches, enabled the identification of iCC’s intratumor heterogeneity and paved the way for the introduction of novel targeted therapies under the scope of precision medicine. Yet, the unmet need for optimal care of patients with chemo-refractory disease or without targetable mutations still exists. Immunotherapy has provided a paradigm shift in cancer care over the past decade. Currently, immunotherapeutic strategies for the management of iCC are under intense research. Intrinsic factors of the tumor, including programmed death-ligand 1 (PD-L1) expression and mismatch repair (MMR) status, are simply the tip of the proverbial iceberg with regard to resistance to immunotherapy. Acknowledging the significance of the tumor microenvironment (TME) in both cancer growth and drug response, we broadly discuss about its diverse immune components. We further review the emerging role of immunotherapy in this rare disease, summarizing the results of completed and ongoing phase I–III clinical trials, expounding current challenges and future directions.

## 1. Introduction

Primary hepatic malignancies, which represent the fourth cause of cancer-related death worldwide, can be mainly classified as hepatocellular carcinoma (HCC) and intrahepatic cholangiocarcinoma (iCC) [1,2]. HCC and iCC are two closely related yet distinct entities that share common molecular and phenotypic characteristics [3]. Interestingly, the combination of HCC and iCC has been also described as a rare type of primary liver cancer, which contains pathologic features of both HCC and iCC cells [4]. Within this context, several clinical trials evaluating novel immunotherapies for patients with primary liver cancer include both HCC and iCC.

The incidence of iCC has been consistently rising in high-income countries, from 0.1 to 0.6 cases per 100,000 over the last three decades [2,5,6], whereas the mortality rates also follow the incidence pattern [7,8]. iCC is defined as a desmoplastic stroma-rich adenocarcinoma of cholangiocyte origin, arising proximal to the secondary biliary ducts [9,10]. In general, iCC is more prevalent in elderly male patients [11], whilst several risk factors have been identified, including obesity, diabetes, primary sclerosing cholangitis, Caroli’s disease, hepatolithiasis, viral hepatitis, liver fluke infections, and cirrhosis [12,13]. Noteworthily, almost 50% of iCC cases have no identifiable risk factor [14].

Patients with iCC usually present with non-specific symptoms (fatigue, weight loss, abdominal pain, obstructive jaundice) when the disease has been ultimately disseminated [15]. Less than 25% of patients are eligible to undergo surgery, which remains the only curative potential for early-stage disease [16]. Even after microscopically margin-negative (R0) resection, prognosis remains dismal due to the high risk of recurrence; a median overall survival (OS) of 40 months has been reported [17]. There is no clinical evidence at present to support the use of radiotherapy for locoregional cancer control, while the American Society of Clinical Oncology (ASCO) guidelines recommend the addition of adjuvant capecitabine [18], based on the reported benefit in both relapse-free survival (RFS) and OS of the randomized phase III BILCAP trial [19].

Current treatment options for advanced-stage iCC are limited and the median OS of these patients varies between 2.5 and 4.5 months [20,21]. Since 2010, the combination of gemcitabine and cisplatin represents the standard of care for patients with locally advanced or metastatic disease, based on the results of the landmark phase III ABC-02 study and the phase II BT22 trial [22,23]. Recently, gemcitabine plus tegafur/gimeracil/oteracil (TS-1) demonstrated non-inferiority to the aforementioned doublet regimen, with acceptable tolerability, in Japanese patients with advanced biliary tract cancer (BTC) [24]. Nonetheless, more than 50% of patients will inevitably develop disease progression [25]; second-line therapy with modified 5-fluorouracil/folinic acid/oxaliplatin (FOLFOX) and active symptom control resulted in a modest OS improvement, but improved six- and 12-month survival rates, compared to active symptom control alone, according to the ABC-06 trial [26].

The parallel development of omics-based studies and novel targeted therapies have unveiled potential molecular-based treatment options for chemo-refractory disease. Indeed, more than 60% of cholangiocarcinoma patients harbour unique gene aberrations including fibroblast growth factor receptor (FGFR) 2 gene translocations, isocitrate dehydrogenase-1 (IDH1) and KRAS proto-oncogene mutations, and receptor tyrosine-protein kinase erbB-2 (ERBB2) amplification [27,28,29]. Pemigatinib is a selective, oral FGFR1-3 inhibitor and the first United States Food and Drug Administration (FDA) and European Medicines Agency (EMA) approved targeted agent for the second-line treatment of iCC [30,31,32]. Even though neurotrophic tyrosine receptor kinase (NTRK) rearrangements are found in less than 5% of BTC, they represent targetable alterations; both larotrectinib and entrectinib are FDA approved tissue agnostic drugs for NTRK fusion-positive tumors [33]. Moreover, dabrafenib plus trametinib showed promising antitumor activity in BRAF V600E-mutated BTC in the phase II, basket trial, ROAR [34], while the combination of bevacizumab and erlotinib may constitute an alternative therapeutic option for patients with epidermal growth factor receptor (EGFR)-mutated advanced BTC [35]. The compelling results of the phase III ClarIDHy trial, regarding the use of ivosidenib in IDH-mutated cholangiocarcinoma, have been recently presented at the 2021 Gastrointestinal Cancers Symposium [36].

Despite these emerging advances towards an individualized treatment plan for iCC, there still exists a paucity of efficacious therapeutic options for this highly challenging and biological heterogeneous malignancy. In recent years, immune-oncology has revolutionized the therapeutic arsenal of various solid tumors, yet its efficacy in BTC merits further investigation [37,38]. Herein, we summarize and critically discuss current evidence, challenges, and future perspectives with regards to the emerging role of immunotherapy in iCC.

## 2. Immunological Characterisation of iCC

Immunotherapy, consisting of (a) immune checkpoint inhibitors (ICIs) targeting programmed death 1 (PD-1), programmed death-ligand 1 (PD-L1), and cytotoxic T-lymphocyte antigen-4 (CTLA-4), (b) cancer vaccines, and (c) adoptive cell transfer (ACT), has blossomed over the last decade, leading to a dramatic advancement of cancer therapeutics [39]. Thoroughly, immune-oncology is emerging as the fifth pillar of cancer treatment, alongside surgery, radiation therapy, chemotherapy, and targeted therapies; yet its efficacy varies and only a subset of patients (approximately 10–35%) achieves durable responses [40]. Taking into account that immunotherapy aims to enhance natural, anti-tumor immune responses, including both innate cells (neutrophils, macrophages, dendritic cells, natural killers, etc.) and adaptive cells (T- and B- lymphocytes) infiltrations into the tumor microenvironment (TME) [41], it becomes prominent that the improved cellular and functional characterization of the immune landscape within TME holds tremendous clinical potential, in both predicting responsiveness to immunotherapy and in identifying novel treatment strategies.

Hence, the TME is a highly dynamic, tridimensional, sophisticated interplay comprising of cancer, stromal, and endothelial cells, which encompasses as well as an abundance of immune components, proliferative factors, vasculature, nerve fibers, extracellular matrix, and acellular components [42]. According to their cytotoxic T cell infiltration, solid tumors are classified into four distinct phenotypes, namely hot, altered immunosuppressed, altered excluded, and cold, with the first exerting the greatest benefit from ICI-based therapy and the latter representing the most challenging to treat [43,44].

Immunotherapy approaches have emerged as viable therapeutic options for iCC, albeit available data are limited to sub-analyses of either small single-arm studies or basket trials; the conflicting results, so far, highlight the unmet need for discovery and validation of predictive biomarkers [45]. In a transcriptome study which categorized 566 cases of iCC, based on their cellular TME features, 11% displayed hot phenotype, whereas 45% showed an immune desert—cold phenotype [46]. Furthermore, PD-L1-expressing iCC cells, which derived from the preinvasive intraductal papillary neoplasms of the bile duct (IPNB), have been associated with PD-L1+ mononuclear cells, PD-1+ lymphocytes, and cytotoxic CD8+ lymphocytes infiltrations within the tumor, thus with PD-1/PD-L1 axis activation, suggesting ICI efficacy [47,48].

Given that the cross-talk between tumor cells and TME, along with genetic and epigenetic alterations, dictate cholangiocarcinoma’s phenotype [49], it is of paramount importance to facilitate an in-depth knowledge of iCC’s immune contexture in order to unravel potential pathophysiological mechanisms behind these controversial responses to immunotherapy. iCC’s prominent histological hallmark is its rich desmoplastic stroma containing primarily cancer-associated fibroblasts (CAFs) and an exceedingly reactive TME of (a) innate immune cells with mainly immunosuppressive and pro-tumorigenic function, like tumor-associated macrophages (TAMs), tumor-associated neutrophils (TANs), and dendritic cells (DCs), and (b) adaptive immune cells, including tumor infiltrating lymphocytes (TILs) and natural killers (NKs) [8,50,51,52]. This complex, multicellular compartment is orchestrated by various soluble signaling mediators (chemokines, extracellular vesicles) and growth factors like platelet-derived growth factor (PDGF) [52] and contributes to iCC’s pathogenesis [8,50,51] (Figure 1).

### 2.1. Cellular Components of TME

#### 2.1.1. Cancer-Associated Fibroblasts (CAFs)

CAFs represent one of the most prominent and critical cellular components of iCC’s desmoplastic stroma [52], which originate from diverse cell populations. Potential sources of CAFs include hepatic stellate cells and portal fibroblasts, bone marrow-derived mesenchymal stem cells (MSCs), cholangiocyte-derived fibroblasts through epithelial-to-mesenchymal transition (EMT), and endothelial cells through endothelial-to-mesenchymal transition (EndMT) [52,53,54,55,56,57,58,59,60,61].

Apart from the direct tumor-promoting activities (cancer cell proliferation, survival, and migration), CAFs also mediate angiogenesis, lymphangiogenesis, and tumor-promoting inflammation via secretion of growth factors like vascular endothelial growth factor (VEGF), fibroblast growth factor (FGF), and soluble chemokines such as CXCL2, CXCL12, CXCL14, interleukin 8 (IL-8), IL-13, stromal derived factor -1 alpha (SDF-1a), and monocyte chemotactic factor 1(MCP-1) [7,56]. Moreover, CAFs elicit structural remodeling of the extracellular matrix (ECM), by the release of protein-lysine 6-oxidase (LOX), matrix metalloproteases (MMPs) like MMP1, MMP2, MMP3, MMP9, and matricellular proteins including periostin, tenascin-C, and osteopontin, thus promoting tumor growth and local invasiveness [7]. The modified, stiffer ECM serves as a barrier itself to immune cells’ invasion. It has been demonstrated that the abundant a-smooth muscle actin (a-SMA) + CAFs correlate with decreased cytotoxic T cell infiltration, improved tumor-associated macrophages (TAMs) infiltration, and poor clinical outcome [7,53,62,63]. Moreover, in a transcriptomic analysis of 10 iCCs, high VEGF expression was associated with TME polarization, restricted T cell infiltration, and poor responsiveness to immunotherapy [64].

Lastly, CAFs display immunosuppressive properties; CAFs’ pro-inflammatory role, when overexpressing fibroblast activation protein (FAP), could be sustained by CCL2-STAT3 signaling, resulting in inhibited myeloid-derived suppressor cells (MDSCs) infiltration in murine models of cholangiocarcinoma [65], whereas the interaction between CAFs and dendritic cells (DCs) dampens the activation of tumor-infiltrating lymphocytes (TILs) [66].

#### 2.1.2. Tumor-Associated Macrophages (TAMs)

TAMs, and especially macrophages with M2 phenotype and high phagocytosis capacity, constitute the key immune cell population infiltrating TME [67,68]. Ontogenetically, hepatic macrophages can be classified as resident Kupffer cells and as recruited bone marrow-derived (or monocyte-derived) hepatic macrophages, yet their exact origin remains to be elucidated [69,70]. In iCC, the increased levels of tumor necrosis factor- alpha (TNF-α) expressed by Kupffer liver cells promotes abnormal cell proliferation and carcinogenesis, through the c-Jun N-terminal kinase (JNK) pathways activation [70,71]. The recruitment of bone marrow-derived monocytes, which differentiate into TAMs, is driven by circulating mediators released by both tumor and stromal cells, including cytokines (IL-1β, IL-10, IL-13, IL-34), osteoactivin, VEGF-A, and colony stimulating factor-1 (CSF-1) [68,72,73].

The close cross-talk between TAMs and cholangiocarcinoma cells generates a plethora of tumor-promoting phenomena. First, TAMs can exert angiogenesis via secreting pro-angiogenic factors (VEGF-A, IL-8, angiopoietin) and pro-inflammatory mediators like cyclooxygenase-2 (COX-2) and inducible nitric oxide synthase (iNOS) [74]. Second, TAMs play a pivotal role in cholangiocarcinoma cell proliferation via the activation of the Wnt/β-caterin signaling pathway [75,76]. Third, TAMs contribute to T cell immunosuppression, promoting tumor growth and metastasis through the expression of hypoxia-inducible factor 1-alpha (HIF-1α) [77].

Despite, CD68+ TAMs have been correlated with a reduced risk of iCC recurrence [78], yet high density of TAMs in cholangiocarcinoma tissues associates with poor prognosis [79].

#### 2.1.3. Tumor-Associated Neutrophils (TANs)

Intratumoral neutrophils (TANs) are engaged into the TME through the angiogenic chemokine CXCL5, which is secreted by tumor and stroma cells via the activated phosphatidylinositol 3-kinase (PI3K)/protein kinase B (AKT) and extracellular signal-regulated kinase 1/2 (ERK1/2) pathway [80]. The presence of increased circulating neutrophil-to lymphocyte ratio (NLR) has been associated with poor clinical outcome in iCC patients [81,82], whilst CD66+ or CD15+ TANs in cholangiocarcinoma have also been correlated with worse survival [83,84,85]. TANs’ role in cholangiocarcinoma remains indefinite and merits further investigation, given their recently apparent importance in cancer progression. Indeed, neutrophil-based therapeutics is likely to become a rapidly growing field of research [86].

#### 2.1.4. Natural Killers (NKs)

NKs, a CD3-CD56+ subset of innate lymphoid cells, account for 30–40% of liver lymphocytes [87] and are characterized by their natural cytotoxic capacity to recognize and destroy malignant cells, in an antigen nonspecific way, via cell lysis [88,89]. Besides inhibiting cell proliferation and metastasis, NKs produce cytokines (mainly interferon-γ, INF-γ), resulting in adaptive immunity modulation [90,91] and decreasing off-target effects through specific antitumor cytotoxic activity [92].

In iCC, endogenous CXCL9 expression has been closely correlated with high NKs infiltrations [93], which in turn inhibit tumor growth and reduce chemoresistance [89,94]. Jung et al., demonstrated that the infusions of human NKs in cholangiocarcinoma cell line HuCCT-1-derived xenograft mouse models induced significant tumor growth inhibition, thus highlighting the potential of NKs-based immunotherapy [89].

#### 2.1.5. Tumor-Infiltrating Lymphocytes (TILs)

TILs comprise of highly heterogeneous immune cells, including CD8+ T cells (cytotoxic T lymphocytes, CTLs), CD4+ T cells (T helper lymphocytes, Th cells), CD20+ B lymphocytes, and CD4+CD25+ regulatory T cells (Tregs) [95]. Immunohistochemical staining of cholangiocarcinoma specimens reveals variable localization of distinct subsets of TILs within the tumor; CD8+ TILs prevail in the intratumoral region, whereas CD4+ T cells are prevalent across the peritumoral area [15,96]. CD8+ TILs preferentially accumulate in loose fibronectin and collagen peritumoral stroma and migrate poorly through dense tumor tissue, guided by chemokines, since they are not capable of producing ECM-degrading proteases [97,98]. Thus, the distribution of CD8+ lymphocytes in TME could be dictated by both density and orientation of a stroma’s structural components.

TILs confer prognostic value in BTC, as well as in other malignancies [99,100]; high numbers of CD8+ and CD4+ TILs within TME correlate with better OS, while low frequencies of CD8+ lymphocytes are associated with unfavorable prognosis [15,83,85,94,99,100,101,102,103]. As aforementioned, an elevated NRL has been related to poor outcomes in iCC [81,82] and has been correlated with higher percentage of IFN-γ+ TILs, CD8+ TILs, and PD-1+CD4+ TILs [104]. Notably, the expression of PD-L1 is more prominent on immune cells rather than on tumor cells [105,106,107,108]. In fact, PD-L1 expression on cancer cells associates with lower responses to immunotherapy [109].

Furthermore, various tumor-associated antigens (TAAs) have been identified in cholangiocarcinoma and represent attractive candidates for cancer vaccines [110]. In a recent transcriptomic study of 10 iCC cases, which revealed the variability among T cell populations within TME, CD8+ TILs from highly heterogeneous tumors exerted decreased cytotoxic capacity [64].

With regards to B TILs, their pathogenic relevance in cholangiocarcinoma has not yet been clarified, yet higher CD20+ B lymphocytes populations have been observed in a rare type of Epstein-Barr virus (EBV) infection-related iCC [111]. Moreover, elevated numbers of CD20+ B TILs have been reported in low-grade tumors and are associated with favorable prognosis [15,96].

Forkhead box P3 (FOXP3), also known as scurfin, is a transcription factor overexpressed by Tregs and associates with the up-regulation of CTLA-4 expression, which in case of iCC is predictive of tumor recurrence and chemoresistance [112]. Additionally, in the presence of CCL2 produced by CAFs, TAMs, and tumor cells, cytotoxic TILs acquire a CD4+CD25+ Tregs phenotype, which leads to secretion of the immunosuppressive transforming growth factor-beta (TGF-β) and IL-10 [113].

#### 2.1.6. Dendritic Cells (DCs)

Along with macrophages and B cells, DCs are among the most important antigen-presenting cells (APCs), capable of providing the necessary signals for effective T cell activation into peripheral lymphoid organs [114]. Whereas immature CD1a+ DCs are found into the tumor, mature CD83+ DCs reside mainly into the invasive front [103]. In cholangiocarcinoma, the suppression of IL-10 and TGF-β receptors on self-differentiated DCs generated intensified activation of effector T cells against cancer cells [115]. In another study, the decreased number of TNF-α producing DCs in cholangiocarcinoma patients could affect DC-mediated immunity [116].

#### 2.1.7. Myeloid-Derived Suppressor Cells (MDSCs)

MDSCs represent a major obstacle for natural antitumor immunity, as a heterogeneous group of immature myeloid progenitors which suppresses both the innate and adaptive immune responses [117]. Recruited mainly by CAFs to the TME, MDSCs exhibit suppressor activity, especially upon CD8+ T cells, by synthesizing reactive oxygen species (ROS) and by affecting the metabolism of L-arginine [65,118,119]. Extensive research regarding their role in iCC pathogenesis may permit the future development of novel immunotherapeutic agents.

### 2.2. Non-Cellular Components of TME

#### 2.2.1. Extracellular Matrix (ECM)

Intense structural remodeling of the ECM through the release of MMPs, periostin, tenascin-C, and osteopontin is crucial in cholangiocarcinogenesis, as described above [7]. Indeed, the overexpression of matricellular proteins has been related to tumor growth, lymph node metastatic potential, and poor OS [120,121]. Elevated expression of osteopontin in stromal cells, in particular, has been significantly correlated with tumor size, local and distant invasion, and advanced stage, through the activation of RAS-RAF-MEK-ERK and Wnt/β-caterin signaling pathways, in BTC patients [122,123]. Moreover, osteopontin promotes NKs’ development and T cells’ survival [124,125]. Further studies will focus on the ECM therapeutic potentials in cholangiocarcinoma, including iCC.

#### 2.2.2. Extracellular Vesicles (EVs)

EVs represent small membranous structures secreted from cells, serving as intercellular messengers [126]. They can be classified as (a) exosomes with 30–100 nm diameter, (b) microvesicles with 0.1–1 μm diameter, and (c) apoptotic bodies with >1 μm diameter which are more commonly eliminated by phagocytes immediately after their release from apoptotic cells [127]. Their components include proteins, lipids, messenger ribonucleic acids (mRNAs), microRNAs (miRNAs), long non-coding RNAs (lncRNAs), and circular RNAs (circRNAs), which are protected from enzymatic degradation by their lipid bilayer [128].

Several miRNAs ad lncRNAs have been identified as candidates in cholangiocarcinoma tissues [129,130]. Previous study demonstrated that miRNA-15a downregulation in cholangiocarcinoma-associated CAFs promotes increased secretion of plasminogen activator inhibitor-2 (PAI-2), which in turn leads to the migration of cholangiocarcinoma cells [131]. The therapeutic significance of miRNAs has been indicated in rat models, whereas the administration of miRNA-195- enriched EVs decreased tumor size and led to increased survival rates [132].

#### 2.2.3. Chemokines

Chemokines are a class of cytokines with approximately 8–12 kilodaltons mass and the ability to induce chemotaxis of nearby leukocytes (monocytes, macrophages, T lymphocytes, etc.) [133]. Together with their receptors, they represent emerging key factors of a well-orchestrated network of tumor-promoting events, including cancer cell migration, invasion, and immune evasion [133].

CXCL1–3, CXCL5–6, CXCL8, CXCL12, CCL2, CCL11, and CCL16 are among the various chemokines which contribute to angiogenesis, either in a direct or indirect way [134]. In cholangiocarcinoma, CXCL12 is positively modulated by angiotensin II and negatively by TGF-β, while CXCR4 expression is promoted by TNF-a-secreting TAMs [133]. Interactions between CXCR4 and CXCL12 have been demonstrated to activate the PI3K/AKT and ERK1/2 pathway, and Wnt/β-caterin signaling pathway [133]. Via the activation of the PI3K/AKT and ERK1/2 pathway, CXCL5 acts as chemoattractant for neutrophils, while FAP-induced CCL2 secretion mediates the migration of both macrophages and MDSCs [133]. Ongoing trials aim to assess the clinical utility of chemokines in patients with hematological malignancies and breast cancer [135,136].

#### 2.2.4. PD-L1, PD-1, and CTLA-4 Expression

Manipulations of the immune checkpoints PD-1 and CTLA-4 are among the immune escape mechanisms of cancer cells and the PD-1/PD-L1 pathway seems to play a pivotal role in the development of a tumor-tolerant TME in BTC [137,138]. Indeed, high tumor cells expression of PD-L1 and PD-1, accompanied by decreased CD3+ and/or CD8+ infiltrates, associates with worse clinical outcome [99,112,139]. Nakamura et al. demonstrated an upregulation of the expression of immune checkpoint molecules in approximately 45% of BTC, which was indicative of worst prognosis [140]. Similarly, Gani et al., showed that 72% of iCC samples expressed PD-L1, which was associated with a 60%-reduction in OS, compared with negative tumor tissue samples [108]. In another study, PD-1 and PD-L1 were expressed in a total of surgically resected iCC, with PD-1 expressed only on TILs and PD-L1 expressed on 30% of tumor cells, TILS, and macrophages [141].

Moreover, FOXP3 overexpression by cholangiocarcinoma cells correlates with lymph node metastasis and poor survival [83,142]. Alongside, FOXP3 overexpression often occurs in association with CTLA-4 overexpression, which holds unfavorable prognostic value in cholangiocarcinoma [112]. Downregulation of the major histocompatibility complex-I (MHC-I) might also play a crucial role in tumor escape from immunosurveillance, and has been related to decreased TILs infiltration and poor prognosis in BTC [15].

## 3. Immunotherapeutic Strategies

Since 2010, immunotherapy has heralded a new era in cancer treatment. On the basis of limited therapeutic options in patients with BTC, including iCC, immunotherapeutic strategies with checkpoint inhibitors, peptide- and dendritic- based vaccines, and adoptive cell therapy, alone or in combination with targeted therapy and/or chemotherapy, have been in progress. Table 1 summarizes results from recently completed immunotherapy trials in BTC patients, including iCC, whereas Table 2 and Table 3 present ongoing trials of immunotherapeutic strategies for the treatment of BTC.

### 3.1. Immune Checkpoint Inhibitors

CTLA-4 and PD-1 are both immune checkpoints expressed by activated T cells, which could dampen the immune response by downregulating the function of activated T cells, in order to maintain immunologic balance. Tumor cells enhance their survival potential tolerance by stimulating this negative signaling. ICIs aim to block this inhibitory signaling, thus restore immune system’s function to destroy cancer cells [143]. Ipilimumab, an anti-CTLA-4 monoclonal antibody (MAb), was the first ICI to gain FDA approval, for the treatment of melanoma, in 2011 [144]. Since then, several ICIs have been approved for cancer therapy, including the anti-PD-1 MAbs nivolumab, pembrolizumab, and cemiplimab, and the anti-PD-L1 MAbs atezolizumab, avelumab, and durvalumab [145], whereas the discovery of CTLA-4 and PD-1 blockade by James P. Allison and Tasuku Honjo, respectively, led them to the Nobel-Prize in 2018 [146].

In the case of approved ICIs for BTC, the phase II basket trial of pembrolizumab monotherapy, which demonstrated significant OS benefit and improved objective response rate (ORR) in mismatch repair (MMR)-deficient patients and led to the FDA approval of pembrolizumab for patients with inoperable or metastatic solid tumors with MMR deficiency or high microsatellite instability (MSI), included four patients with ampullary cancer or cholangiocarcinoma [147,148]. Notably, Lynch syndrome, known as hereditary non-polyposis colorectal cancer (HNPCC), is characterized by MMR deficiency and represents, despite uncommon (almost 2% of cases), a risk factor for iCC [149].

As aforementioned, PD-1 and PD-L1 are expressed in iCC tissue samples, underlying the therapeutic potential of targeting the PD-1/PD-L1 pathway [107,150,151]. Furthermore, there is growing evidence that ICIs demonstrate promising clinical efficacy in virus-associated cancers, like Hodgkin lymphoma, merkel cell carcinoma, head and neck-, and hepatocellular- cancer, probably due to the increased neo-antigen presentation [152]. Noteworthy, various oncogenic viruses have been related to iCC’s development [12,13]. Currently, several phase I-III clinical trials are underway, using ICIs alone or in combination with other chemotherapeutic modalities or targeted therapies.

The phase Ib KEYNOTE-028 and phase II KEYNOTE-158 trials, whereas pembrolizumab resulted in durable, high response rates of 40% in previously treated, advanced non-colorectal, MSI-high/MMR-deficient cancer patients, included 22 (9.4%) cholangiocarcinoma cases and retrospectively evaluated PD-L1 status [153]. Indeed, PD-L1 expression, which was defined as a combined positive score of at least 1% of tumor and inflammatory (lymphocytes, macrophages) cells, was noted in all cholangiocarcinoma tissue samples [154]. In the majority of cases, treatment-related adverse events were mild to moderate regarding severity and occurred in almost 65% of patients, while 6% of patients experienced grade 3 immune-mediated adverse events or infusion reactions [153].

Pembrolizumab demonstrated promising antitumor activity when combined with lenvatinib, an oral multiple kinase inhibitor against the VEGFR1, VEGFR2, and VEGFR3 kinases, in the ongoing phase II LEAP-005 study, which evaluated both the efficacy and safety of this regimen, among others, in 31 previously treated, advanced BTC patients [155].

A multicenter phase II trial demonstrated durable partial response in MMR-proficient, refractory BTC patients treated with nivolumab as second-line therapy. 32 (59%) patients had iCC, while PD-L1 expression predicted response to immunotherapy and correlated with prolonged progression-free survival (PFS) [156]. In the case of a 40-year-old female patient with recurrent iCC, the combination of nivolumab and lenvatinib showed promising efficacy, with the complete response in liver metastases and the stabilization of lung lesions within nine months [157]. Nivolumab has also been examined in the first-line setting, whereas its combination with gemcitabine and cisplatin resulted in a RR of 37% and a median OS of 15.4 months [158]. Durable responses have also been reported with the combination of nivolumab and ipilimumab in patients with advanced iCC, irrespective of biomarkers status, according to a subgroup analysis of the phase II CA209-538 trial [159].

The combination of chemotherapy and durvalumab, alone or in addition to tremelimumab (anti-CTLA-4 antibody), apart from being tolerable, resulted in high response- and disease control- rates in the majority of chemotherapy-naïve advanced BTC patients, in a phase II study [160]. Currently, the combination of gemcitabine and cisplatin with durvalumab is being under further investigation in the phase III TOPAZ-1 trial (NCT03875235).

M7824, also known as bintrafusp alfa, is a first-in-class bifunctional fusion protein immunotherapy, which combines a TGF-β ‘trap’ with an anti-PD-L1 immunoglobulin G (IgG) monoclonal antibody, thus simultaneously targeting two immune-suppressive pathways in the TME [161]. Data from an ongoing expansion cohort from a phase I trial (NCT02699515), with 30 patients with BTC who progressed after platinum-based first-line therapy, highlight its therapeutic potential [162], which is at present under evaluation either as monotherapy (NCT03833661) or in combination with platinum-based chemotherapy (NCT04066491).

Poly adenosine diphosphate-ribose polymerase inhibitors (PARPi) represent a novel class of targeted cancer therapy, by exploiting synthetic lethality in tumours which harbour germline and somatic alterations in DNA damage repair (DDR) genes; whereas their combination with immune-oncology appears to be a promising treatment approach for solid tumors [163]. The biological rationale behind their synergistic effects has been thoroughly determined; DDR gene aberrations have been associated with genomic instability, immunomodulation of TME, and increased tumor mutational burden (TMB) [164,165,166,167,168,169]. DDR mutations have been observed in 28.9–63.5% of BTC, with BRCA mutations occur-ring in less than 7% of these patients [170,171,172]. Currently, two early-phase trials (NCT03639935, NCT03991832) are underway to evaluate the efficacy from the combination of PARPi and ICI treatment and uncover the patient population which will potentially derive the greatest benefit from this combo.

### 3.2. Cancer Vaccines

TAAs, normally expressed in tumors, represent theoretically ideal targets for cancer vaccines, since they allow specific T cell responses. However, loss or downregulation of major histocompatibility complex class I molecules (MHC-I or HLA-I), by which tumor antigens are presented to CTLs, are among the T cell immunosurveillance mechanisms [173]. In iCC, derangements in HLA-I have been associated with more advanced stages, despite the underlying mechanism has not yet been elucidated [141]. Various TAAs, like Wilms’ tumor gene 1 (WT1) and mucin 1 (MUC1), have been studied as potential targets in BTC patients [174,175,176,177,178]. Both of them have been reported in approximately 80–85% of iCC cases and are of negative prognostic value [177,178].

In a Japanese open-label, dose-escalation, phase I trial, which enrolled 4 patients with iCC, the combination of WT1 peptide vaccine and gemcitabine resulted in a median OS of 9.5 months, with good tolerability [179]. The researchers now investigate in a randomized phase II study the combination of WT1 vaccine with gemcitabine and cisplatin as first-line treatment in patients with unresectable or recurrent BTC [180].

Yamamoto et al., had accrued patients with mixed gastrointestinal tract cancers (pancreatic and BTC) in a phase I trial of vaccination with MUC1 peptides and incomplete Freund’s adjuvant (Montanide ISA51), which failed to provide survival benefit [181]. Similarly, Lepisto et al. conducted a phase I/II trial of vaccination with DCs loaded with MUC1, as adjuvant therapy in 10 patients with early-stage pancreatic cancer and two with stage II iCC. One of the two iCC patients remained free of recurrence [182]. These small-scale trials proved the satisfactory profile of the vaccines, even if the survival data were rather inconclusive.

Promising results have been obtained by a case report of a female patient with advanced iCC, repeatedly operated for disease recurrences, whereas her immunization with a personalized multi-peptide vaccine, targeting tumor mutations presented by HLA-I, conferred durable response [183]. Another phase II trial of personalized peptide vaccine evaluated the feasibility of HLA-matched vaccine peptides in six iCC patients and proved that low levels of IL-6 were significantly associated with improved OS (hazard ratio = 1.123; 95% CI 1.008–1.252; *p* = 0.035) [184]. Based on this result, the same group had started an early phase trial to examine whether blockade of IL-6-mediated inflammation with tocilizumab could enhance the immune responses after personalized peptide vaccination in advanced BTC patients [185,186].

In a single-arm phase II trial, combination of gemcitabine plus elpamotide (an HLA-A* 24:02-restricted epitope peptide of VEGFR-2, which induces CTL responses) in 54 patients with unresectable or recurrent BTC demonstrated moderate antitumor effect, response rate of 18.5%, and median OS of 10.1 months in comparison with the 7.6 months of the historical control [187].

Tumor lysate-based DCs cancer vaccines remain under clinical investigation, yet early efficacy has been demonstrated in in-vitro studies [188]. In an early phase trial, the autologous tumor lysate-pulsed DCs injection and the following transfer of CD3-activated T cells in postoperative patients with iCC led to improved PFS and OS, compared to the control group who received surgery alone (18.3 versus 7.7 months; *p* = 0.005 and 31.9 versus 17.4 months; *p* = 0.022, respectively) [189]. We still await the results of two recently completed phase I clinical trials for DC-based immunotherapy in BTC (NCT00027534, NCT00004604).

### 3.3. Adoptive Cell Therapy (ACT)

Adoptive cell therapy with (a) autologous TILs infusion or (b) genetically modified T cells to express either T cell receptor (TCR) or chimeric antigen receptor (CAR) represent another therapeutic strategy to manipulate the immune system to recognize and destroy tumor cells, with undeniable results seen in melanoma patients [190]. In the case of BTC, ACT has been limited to small case-series, case reports, and single-arm early phase studies.

Back in 2006, a patient with operable, node-positive iCC received adjuvant CD3-activated T cells and tumor lysate- or peptide-pulsed DCs, and survived 3.5 years [191]. ACT of TILs containing CD4+ erbb2 interacting protein (ERBB2IP) mutation-reactive T cells generated promising results in a patient with metastatic cholangiocarcinoma; the patient achieved a 30% decrease in tumor lesions, with prolonged disease stabilization of 13 months, while the rechallenge with a second T cell infusion resulted in further response [192]. Recently, a phase I trial demonstrated encouraging results of clinical activity of CAR-T cell immunotherapy targeting human epidermal growth factor receptor 2 (HER2) in 11 patients with advanced BTC and pancreatic cancers [193].

## 4. Conclusions

The role of immunotherapy in iCC is currently investigational and the results of ongoing studies are eagerly anticipated. Based on promising results from several solid tumors clinical trials, it will hopefully increment patient outcomes in cases of advanced disease. This rapidly transformational landscape of immune-oncology in cholangiocarcinoma poses several challenges. In order to consider this advancing therapeutic strategy in the exquisitely heterogeneous iCC, a deeper, rudimentary, understanding of the cellular diversity of its constantly evolving TME is fairly crucial. The milieu of tumor and immune cells, along with vasculature, extracellular matrix, and signaling molecules regulate immune responses and influence immunotherapy’s efficacy. The modulation of the crosstalk between iCC and TME, by targeting the activation of these cells, in combination with ICIs represents an attractive therapeutic prospective [194,195].

The next frontier of this emerging field is the development of combinational therapeutic approaches. Such approaches, constituting of ICIs along with chemotherapy and/or targeted therapy, have garnered clinical research interest, as they represent appealing strategies for enhanced efficacy, by overcoming both primary and acquired resistance. The rational design of preclinical research to benchmark the drug combinations, side-by-side, against monotherapies and/or other combinations will confer important insights into the degree of synergy [196]. Last but not least, the lack of validated predictive biomarkers, in order to determine the patients’ sub-cohorts which will derive benefit from immunotherapy, represents a crucial challenge and should be underscored. To date, no single biomarker, including TME, MMR/MSI status, TMB, and PD-L1, has proved efficient to predict response to immune-targeted therapies and guide treatment decisions in iCC clinical settings [45].

Though representing a relatively young field, immunotherapy in iCC is burgeoning, with several gaps that need to be filled. It is of the utmost importance to fully decipher the underpinnings of the optimal antitumor immune response, at both cellular and molecular levels, in the effort to optimize its efficacy against this challenging malignancy.

## Figures and Tables

**Figure 1 vaccines-09-00422-f001:**
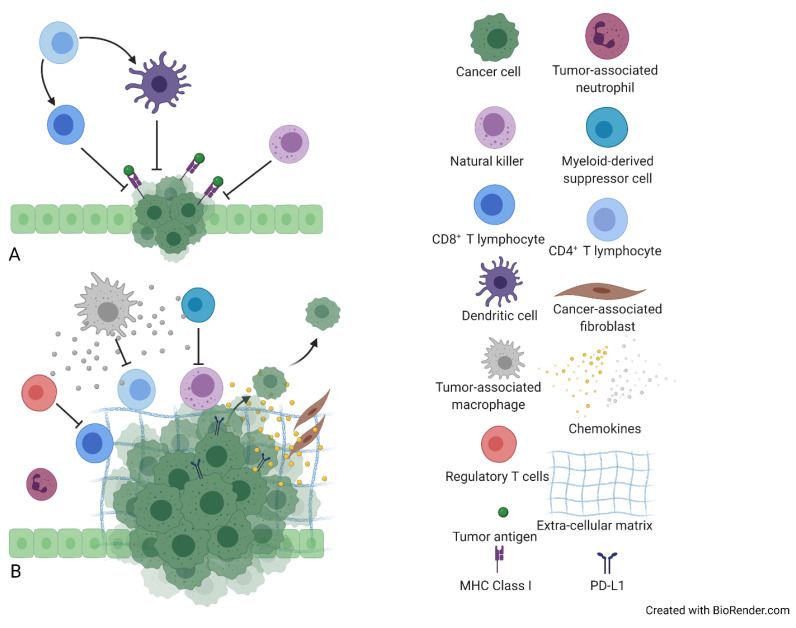
Immunological features of intrahepatic cholangiocarcinoma (iCC) tumor microenvironment (TME). (**A**) The cross-talk between anti-tumorigenic immune cells, including tumor-infiltrating lymphocytes (TILs), dendritic cells (DCs), and natural killer cells (NKs) and tumor cells. (**B**) Cancer-associated fibroblasts (CAFs) and tumor-associated macrophages (TAMs) contribute to tumorigenesis, by secreting a plethora of growth factors, proteases, and cytokines. Responding to the activated chemokine axes, myeloid-derived suppressor cells (MDSCs) and regulatory T-cells (Tregs) are recruited into TME, leading to immune surveillance disruption. Moreover, extracellular matrix (ECM) has been correlated with tumor growth, whereas programmed death-ligand 1 (PD-L1) expression mediates tumor immune escape.

**Table 1 vaccines-09-00422-t001:** Reported outcomes of immunotherapy for biliary tract cancer including intrahepatic cholangiocarcinoma.

Clinical Trial Identifier	Phase	Setting	Regimen	Target	No. of Patients	Outcomes
NCT02054806	Ib	Second-, or later-line	Pembrolizumab	PD-1	24	mPFS of 1.8 months; mOS of 5.6 months; ORR of 13%; SD rate of 17%
NCT02628067	II	Second-, or later-line	Pembrolizumab	PD-1	104	mPFS of 2.0 months; mOS of 7.4 months; ORR of 5.8%
JapicCTI-153098	II	Second-, or later-line	Nivolumab	PD-1	30	mPFS of 1.4 months; mOS of 5.2 months; RR of 3%
NCT02829918	II	Second-, or later-line	Nivolumab	PD-1	54	mPFS of 3.7 months; mOS of 14.2 months; ORR of 22%; DCR of 50%
NCT01938612	II	Second-, or later-line	Durvalumab	PD-L1	42	mPFS of 1.5 months; mOS of 8.1 months; RR rate of 4.8%
NCT02699515	I	Second-, or later-line	M7824	PD-L1	30	mOS of 12.7 months; ORR of 20%
NCT03046862	II	First line	Durvalumab + tremelimumab + GEMCIS	PD-L1, CTLA-4	121	mPFS of 11–13 months; mOS of 15–21 months; ORR of 50–73%
NCT02923934	II	First-, or later- line	Nivolumab + ipilimumab	PD-1, CTLA-4	39	mPFS of 2.9 months; mOS of 5.7 months; ORR of 23%
NCT01853618	I	Second-, or later- line	Tremelimumab + RFA	CTLA-4	32	mPFS of 7.4 months; mOS of 12.3 months; PR of 26.3%

CTLA-4: cytotoxic T-lymphocyte-associated protein 4; DCR: disease control rate; GEMCIS: gemcitabine + cisplatin; mOS: median overall survival; mPFS: median progression-free survival; ORR: objective response rate; PD-1: programmed cell death protein 1; PD-L1: programmed death-ligand 1; PR: partial response; RFA: radiofrequency ablation; RR: response rate; SD: stable disease.

**Table 2 vaccines-09-00422-t002:** Ongoing trials of ICC for HCC and biliary tract cancer including intrahepatic cholangiocarcinoma.

Clinical Trial Identifier	Phase	Status	Setting	Regimen	Target	No. of Patients
ICI Monotherapy						
NCT03110328	II	Recruiting	Advanced, refractory BTC	Pembrolizumab	PD-1	33
NCT02054806	Ib	Active, not recruiting	PD-L1 positive cancers, including BTC	Pembrolizumab	PD-1	477
NCT02628067	II	Recruiting	Advanced, refractory cancers, including BTC	Pembrolizumab	PD-1	1350
NCT03695952	Prospective observational cohort	Recruiting	HCC or BTC	Pembrolizumab	PD-1	100
NCT02829918	II	Active, not recruiting	Advanced, refractory BTC	Nivolumab	PD-1	52
NCT02465060	II	Recruiting	Advanced, refractory, MMR-d cancers	Nivolumab	PD-1	6452
NCT03695952	Prospective observational cohort	Recruiting	HCC or BTC	Nivolumab	PD-1	100
NCT03867370	Ib/II	Not yet recruiting	Operable HCC or iCC	Toripalimab	PD-1	20
NCT02091141	II	Active, not recruiting	Advanced, refractory solid tumours, including BTC with high TMB	Atezolizumab	PD-L1	765
NCT02834013	II	Recruiting	Advanced, refractory solid tumours including BTC	Nivolumab + ipilimumab	PD-1, CTLA-1	707
Dual ICIs						
NCT03101566	II	Active, not recruiting	Unresectable, chemo-naïve BTC	Nivolumab + ipilimumab	PD-1, CTLA-1	64
ICI in combination with local ablative therapy						
NCT02821754	II	Recruiting	Unresectable, refractory HCC or BTC	Durvalumab + tremelimumab + TACE, RFA, or cryoablation	PD-1, CTLA-4	90
NCT03898895	II	Recruiting	Unresectable, untreated iCCA, eligible for RT	Pembrolizumab + SBRT vs. GEMCIS chemotherapy	PD-1	184
NCT03482102	II	Recruiting	Unresectable HCC or BTC	Durvalumab + tremelimumab + RT	PD-1, CTLA-4	70
ICI in combination with chemotherapy						
NCT03473574	II	Active, not recruiting	Untreated BTC	Durvalumab + tremelimumab + GEM or GEMCIS vs. GEMCIS chemotherapy	PD-1, CTLA-4	63
NCT03046862	II	Recruiting	Unresectable, untreated BTC	Durvalumab + tremelimumab + GEMCIS chemotherapy	PD-1, CTLA-4	31
NCT03704480	II	Recruiting	Recurrent or advanced, refractory BTC	Durvalumab + tremelimumab + paclitaxel	PD-1, CTLA-4	102
NCT03875235	III	Recruiting	Unresectable, untreated BTC	Durvalumab + GEMCIS vs. GEMCIS chemotherapy	PD-1	474
NCT03257761	Ib	Recruiting	Unresectable, refractory HCC, PDAC, or BTC excluding ampullary	Durvalumab + guadecitabine	PD-1	90
NCT03111732	II	Active, not recruiting	Unresectable, refractory BTC	Pembrolizumab + CAPOX chemotherapy	PD-1	19
NCT03260712	II	Recruiting	Unresectable, untreated BTC	Pembrolizumab + GEMCIS	PD-1	50
NCT03796429	II	Recruiting	Advanced BTC	Toripalimab + gemcitabine	PD-1	40
NCT03101566	II	Active, not recruiting	Unresectable, untreated BTC	Nivolumab + GEMCIS	PD-1	64
NCT03785873	I/II	Recruiting	Unresectable, refractory BTC	Nivolumab + nal-irinotecan + 5-fluorouracil + leucovorin	PD-1	40
NCT03478488	ΙΙΙ	Recruiting	Unresectable, untreated BTC	KN035 + GEMOX vs. GEMOX chemotherapy	PD-L1	390
ICI in combination with targeted therapy						
NCT02393248	I/II	Active, not recruiting	Advanced, refractory solid tumours, including CCA, harboring genetic alteration of FGF or FGFR genes	Pembrolizumab + pemigatinib	PD-1, FGFR 1-3	325
NCT03684811	I/II	Active, not recruiting	Selected solid tumours, including BTC, with IDH1 mutations	Nivolumab + FT-2102	PD-1, IDH1	200
NCT03201458	II	Active, not recruiting	Unresectable, refractory BTC	Atezolizumab + cobimetinib	PD-L1, MEK	82
NCT03639935	II	Recruiting	Advanced, refractory BTC	Nivolumab + rucaparib	PD-1, PARP	35
NCT03991832	II	Recruiting	Selected solid tumours, including BTC, with IDH mutations	Durvalumab + olaparib	PD-1, PARP	59
NCT03829436	I	Recruiting	Advanced solid tumours, including CCA	TPST-1120 + nivolumab, docetaxel chemotherapy or cetuximab	PPARα, PD-1, anti-EGFR	338
NCT03095781	I	Recruiting	Advanced, refractory GI cancers, including CCA	Pembrolizumab + XL 888	PD-1, Hsp90	50
NCT03872947	I	Recruiting	Advanced solid cancers, including CCA	TRK-950 + nivolumab or pembrolizumab or chemotherapy	TAA, PD-1	36
ICI in combination with TME targeted therapy						
NCT02703714	II	Active, not recruiting	Advanced BTC	Sargramostim + pembrolizumab	PD-1	42
NCT03833661	II	Active, not recruiting	Advanced, refractory BTC	M7824	PD-L1/TGFβRII	141
NCT03250273	II	Active, not recruiting	Advanced, untreated CCA or PDAC	Entinostat + nivolumab	PD-1	54
NCT02443324	I	Active, not recruiting	Advanced, refractory cancers, including BTC	Ramucirumab + pembrolizumab	VEGFR-2, PD-1	155
NCT03895970	II	Recruiting	Advanced, refractory, primary liver cancer or BTC	Lenvatinib + pembrolizumab	VEGFR-2, PD-1	50
NCT03797326	II	Recruiting	Advanced, refractory solid tumours, including BTC	Lenvatinib + pembrolizumab	VEGFR-2, PD-1	180
NCT03475953	I/II	Recruiting	Advanced, refractory GI, not MMR-deficient	Avelumab + Regorafenib	PD-L1, RTKs, VEGFR 2-3	212

BTC: biliary tract cancer; CAPOX: capecitabine + oxaliplatin; CCA: cholangiocarcinoma; CTLA-4: cytotoxic T-lymphocyte-associated protein 4; EGFR: epidermal growth factor receptor; FGF: fibroblast growth factor; FGFR: fibroblast growth factor receptor; FOLFOX: folinic acid (leucovorin) + 5-Fluorouracil + oxaliplatin; GEM: gemcitabine; GEMCIS: gemcitabine + cisplatin; GI: gastrointestinal; HCC: hepatocellular carcinoma; iCC: intrahepatic cholangiocarcinoma; ICI: immune checkpoint inhibitor; IDH1: isocitrate dehydrogenase 1; MMR-d: mismatch repair protein deficiency; MUC-1: mucin 1; NK: natural killers; PARP: poly (ADP-ribose) polymerase; PD-1: programmed cell death protein 1; PDAC: pancreatic ductal adenocarcinoma; PD-L1: programmed death-ligand 1; RFA: radiofrequency ablation; RT: radiation therapy; RTKs: receptor tyrosine kinases; SBRT: stereotactic body radiation therapy; TAA: tumor associated antigen; TACE: transarterial chemoembolization; TGFβ: transforming growth factor beta; TMB: tumor mutational burden; TME: tumor microenvironment; VEGFR: vascular endothelial growth factor.

**Table 3 vaccines-09-00422-t003:** Ongoing trials of TME targeted therapy and ACT for HCC and biliary tract cancer including intrahepatic cholangiocarcinoma.

Clinical Trial Identifier	Phase	Status	Setting	Regimen	Target	No. of Patients
TME Targeted Therapy						
NCT03314935	I/II	Active, not recruiting	Advanced solid tumours including BTC	INCB001158 + FOLFOX, GEMCIS or paclitaxel chemotherapy	Arginase, TME	249
NCT03329950	I	Recruiting	Advanced, refractory cancers, including CCA	CDX-1140 +/− CDX-301	CD40, dendritic cell	180
NCT03071757	I	Active, not recruiting	Advanced solid cancers, including CCA	ABBV-368) +/− ABBV-181	OX40, PD-1	170
ACT						
NCT03820310	II	Recruiting	iCC after radical resection with CR	Autologous central memory T cell therapy + radiotherapy or chemotherapy		20
NCT03801083	II	Recruiting	Unresectable, refractory BTC	Autologous TIL		59
NCT03633773	I/II	Recruiting	MUC-1 positive iCCA	Autologous MUC-1 CAR T-cell therapy + fludarabine/cyclophosphamide		9
NCT02482454	II/III	Active, not recruiting	Unresected CCA, without extrahepatic metastasis	Autologous cytokine-induced NK cells + RFA vs. RFA		50

ACT: adoptive cell transfer; BTC: biliary tract cancer; CAR-T cell: chimeric antigen receptor T cell; CCA: cholangiocarcinoma; FOLFOX: folinic acid (leucovorin) + 5-Fluorouracil + oxaliplatin; GEM: gemcitabine; GEMCIS: gemcitabine + cisplatin; HCC: hepatocellular carcinoma; iCC: intrahepatic cholangiocarcinoma; MUC-1: mucin 1; NK: natural killers; RFA: radiofrequency ablation; RT: radiation therapy; RTKs: receptor tyrosine kinases; SBRT: stereotactic body radiation therapy; TME: tumor microenvironment.

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
