# Peer review of "The Emerging Role of Immunotherapy in Intrahepatic Cholangiocarcinoma"

_vaccines, 2021, doi:10.3390/vaccines9050422_

Round 1
Reviewer 1 Report
This is a well written review on the potential role of immunotherapy in intrahepatic cholangiocarcinoma. I have a couple of minor comments:
- For the review on the immunological feature of iCC, the section should be more clearly subdivided into cell and non-cell factors (e.g. cytokines). The last part (immune escape) seems do not fit with general flow. For example, PD-L1 and CTLA-4 expression would be more related to the rest of the headings of the section.
- A figure summarizing the key immunological features of iCC would great aid the readers to understand the whole picture.
Author Response
We would like to thank Reviewer 1 for the positive comments. In response to the 1st suggestion, we indeed subcategorized the tumor microenvironment components into cellular and acellular features, whereas we changed the title of the last part from ‘’immune escape’’ into ‘’PD-L1, PD-1, and CTLA-4 Expression’’, as it was suggested. Moreover, we included a figure summarizing the key immunological features of iCC.
Reviewer 2 Report
This is an excellent survey with regard to the potential emerging role of immunotherapy in intrahepatic cholangiocarcinoma. Therefore, the manuscript should be accepted for publication in its current form under minor revisions.
- Grammatical errors should be corrected throughout the Text.
- Tables should be reduced in size in order to be more helpful.
Author Response
We sincerely appreciate the kind words of Reviewer 2. In order to improve the readability of the manuscript we corrected several grammatical errors throughout the paper, whereas we presented the several ongoing trials in 2 tables.
Reviewer 3 Report
Dear Editor, thank you so much for inviting me to revise this manuscript about the topic of immunotherapy in intrahepatic cholangiocarcinom.
Notably enough, the last decade has witnessed the advent of genomic sequencing, an epochal change that has provided an unprecedented amount of information regarding the processes of cancer pathogenesis in human malignancies. As regards biliary tract cancer, molecular profiling has become increasingly important, with a wide range of studies describing genetic aberrations which are exclusive to specific subtypes of these hepatobiliary tumors; these findings have paved the way towards the development of molecularly targeted therapies in biliary tract cancer, whose role has been explored and is currently under investigation as monotherapy or in combination with other anticancer agents in several phase I to III clinical trials. However, a large proportion of biliary tract cancer patients (approximately 50%) does not harbor potentially actionable aberrations, and the vast majority of data regarding targeted therapies are limited to a highly specific population. In fact, one of our main issues in everyday clinical practice is that “Precision Oncology” is primarily limited to intrahepatic cholangiocarcinoma patients so far, where isocitrate dehydrogenase (IDH) and fibroblast growth factor receptor (FGFR)2 represent the most promising therapeutic targets, as witnessed by several recently published or presented studies.
In this changing, evolving landscape, this paper provides an interesting overview of the current scenario of immunotherapy in intrahepatic cholangiocarcinoma.
On the basis of the above, it addresses a current topic.
The manuscript is quite well written and organized. English could be improved.
The tables are comprehensive and clear.
The introduction explains in a clear and coherent manner the background of this study.
We suggest the following modifications:
- Introduction section: although the authors correctly included important papers in this setting, we believe a couple of studies should be cited within the introduction (PMID: 33382361; PMID: 33215952), only for a matter of consistency. We think it might be useful to introduce the topic of this interesting study.
- We believe some topics should be expanded. Among these, recent reports have suggested a possible interaction between DNA damage repair gene mutations, immunotherapy, microsatellite instability, and tumor mutational burden, and this topic should be added. Notably enough, we are witnessing a growing interest towards PARP inhibitors and the combination of PARPi plus immune checkpoint inhibitors, and more details about this potential therapeutic strategy would be welcome.
We believe this article deserves major revisions. The main strengths of this paper are that it addresses an interesting and very timely question and provides clear answers, with some limitations. We suggest a linguistic revision and the addition of some references for a matter of consistency. Moreover, the authors should better clarify some points and should add some studies, as suggested.
Author Response
For the insightful review of our manuscript, we would like to express our sincere gratitude to Reviewer 3. We had both revised the Introduction, incorporating the suggested citations, and we extended part 3.1. ‘’Immune checkpoint inhibitors’’, emphasizing the promising role of PARP inhibitors in combination with ICIs in iCC.
Round 2
Reviewer 3 Report
The authors addressed all the issues we raised.
We recommend acceptance in its current form.